# Attenuation of Weight Gain and Prevention of Associated Pathologies by Inhibiting SSAO

**DOI:** 10.3390/nu12010184

**Published:** 2020-01-09

**Authors:** Dimitri Papukashvili, Nino Rcheulishvili, Yulin Deng

**Affiliations:** 1School of Life Science, Beijing Institute of Technology, Beijing 100081, China; dimitri@bit.edu.cn (D.P.); nino@bit.edu.cn (N.R.); 2Beijing Key Laboratory for Separation and Analysis in Biomedicine and Pharmaceuticals, Beijing 100081, China

**Keywords:** obesity, semicarbazide-sensitive amine oxidase (SSAO), caffeine, histamine, diabetes

## Abstract

Obesity is a worldwide prevalent metabolic disorder that is associated with diabetes, among many other diseases. Bearing this in mind, prevention and treatment ways need to be improved. Notably, activity of the enzyme semicarbazide-sensitive amine oxidase (SSAO) is found to be elevated in overweight subjects. Moreover, SSAO inhibition has resulted in an increase of histamine activity in adipose tissue and the limitation of body fat. The current review aims to overview the risks of obesity, rationalize the molecular ways of SSAO activity, and outline the strategies of inhibiting upregulated enzyme levels. It describes the differences between SSAO inhibitors and advances the prospective agents. Based on evidence, caffeine is proposed as an effective, safe, and reliable choice to inhibit SSAO activity. Furthermore, the histamine in adipocytes has been associated with SSAO activity. Therefore, it is suggested as one of the key compounds to be studied for obesity management. To conclude, inhibiting SSAO may attenuate weight gain and prevent related diseases.

## 1. Introduction

Obesity is an epidemic complex disorder characterized by excessive body fat which is diagnosed when the body mass index (BMI) is 30 or higher [1]. It is a global problem and plays a pivotal role in increasing the morbidity of many diseases, such as type 2 diabetes mellitus (T2DM), heart diseases, stroke, etc. [1,2,3]. Dependent on the development of world economy, it has become one of the crucial health disorders expanding rapidly [4,5]. Expression and regulation of many peptides and enzymes are impaired in obesity. Hence, a reasonable anti-obesity therapeutic strategy needs to be implemented. It is noteworthy that an enzyme semicarbazide-sensitive amine oxidase (SSAO), also known as vascular adhesion protein-1 (VAP-1), which is responsible for deamination of the primary amines such as methylamine and converts them into cytotoxic aldehydes, e.g., formaldehyde, ammonia, and hydrogen peroxide, is found to be associated with obesity and related diseases. Particularly, the expression and activity of SSAO in the adipose tissue of obese mice are found to be upregulated [6]. Additionally, levels of this enzyme are found to be elevated in overweight subjects and in many obesity related pathologies. Increased serum concentration of SSAO is interconnected with hyperglycemia [7]. It is known that gaining mass in body fat is a major risk for type 2 diabetes mellitus [1,8]. Thus, using the inhibitors of this enzyme can be proposed as a therapy for losing weight. There are numerous compounds studied with the SSAO inhibition capacity. The most common, inexpensive, accessible substance which has a property of SSAO inhibition is caffeine—an alkaloid present in coffee beans and green tea leaves and consumed daily worldwide. There are a number of researches stating that it is helpful for weight loss and maintenance [9,10]. However, these researches focus on the ability of caffeine to boost metabolism and increase energy expenditure [11,12]. In this manuscript, we review another beneficial role of caffeine for overweight subjects, which is the potential to inhibit enzymatic activity of SSAO [13]. Histamine—a naturally occurring bioamine in body and food—merits attention since the activity of this compound is associated with SSAO metabolism and lipolysis in adipose tissue [14].

This manuscript focuses on SSAO inhibitors and, therefore, aids in anti-obesity drug development. Hence, it will be valuable for future researchers working on finding and developing treatments of health complications in the obese population.

## 2. SSAO and Obesity

SSAO is a copper-containing primary amine oxidase [15,16] (PrAO) and its activity is found to be increased in obesity. The SSAO-mediated deamination process in adipocytes seems to be involved in obesity progression [1]. The functional involvement of this enzyme in the fat accumulation process can be affirmed by the facts of copper, copper-containing protein content, and SSAO elevation in the visceral fat of obese patients [17]. Obesity is a common condition for T2DM, as well as other related complications [18]. Moreover, overweight is a typical state for people with prediabetes compared to normal subjects where the enzyme SSAO concentration is also increased. Indeed, this enzyme is associated with glucose transport which is a key process in evolving diabetes disease [3,7]. A number of studies showed that the different levels of H_2_O_2_, which is one of the products of the enzyme in obese subjects, are related with insulin resistance [19,20]. Additionally, SSAO and related metabolic products have been implicated in glucose transportation processes in adipocytes [21,22]. In fact, SSAO activity was found in glucose transporter 4 (GLUT4) vesicles in rat adipocytes, which might indicate that its activity can enhance transportation of GLUT4-containing vesicles on the adipose cell surface where insulin-mimicry occurs [21,23]. Evidently augmentation of SSAO levels in the pancreas organs of diabetic subjects raises insulin secretion [24]. Mercader et al. demonstrated that SSAO substrates such as benzylamine exhibit insulin-like effects in adipose tissue via hydrogen peroxide, which is a metabolic product of SSAO activity and mimics insulin [22,23]. There are two forms of SSAO—soluble and membrane-bound. The levels of the soluble form are physiologically correlated with obesity, inflammation, cardiovascular diseases, and diabetes [25]. Overall, certain researches elucidate that SSAO levels change dependent on adiposity.

## 3. SSAO Inhibitors as Therapeutics for Obesity

Inhibiting SSAO merits an awareness as it can be advantageous for the prevention and treatment of many associated health disorders: cardiovascular diseases [26], diabetes, and its complications such as retinopathy, neuropathy, nephropathy, etc. [27]. As raised SSAO expression and activity are found in obese people, its inhibition is worthwhile. There are numerous publications regarding the effects of SSAO inhibitors [28,29,30,31]. They have been observed to diminish obesity in fat and normal mice and rats [32]. The information about SSAO inhibitors with their feasibility, advantages, and disadvantages is given in Table 1.

Remarkably, the name of the enzyme comes from its sensitivity towards semicarbazide—one of the inhibitors that limit weight gain, fat deposition, and glucose transportation in adipocytes of mice while administered orally [22]. Carpene et al. demonstrated that the combined inhibition of monoamine oxidase (MAO) with an irreversible inhibitor pargyline (dosage of 20 µmol kg^−1^) and SSAO with its already mentioned inhibitor semicarbazide (dosage 36 µmol kg^−1^) can reduce fat deposition [32]. SSAO is highly sensitive to semicarbazide [30,41], which significantly suppresses the body weight gain in Wistar Hannover GALAS rats of both sexes [42], however, severe health outcomes followed due to the chronic toxicity and carcinogenicity of this inhibitor [43]. PXS-4681A has shown a noteworthy irreversible inhibition property on mice [35] and rats [34]. The mentioned compound represents a very suitable candidate for clinical progression, which is a mechanism-based inhibitor with lasting and lower dosing at 2 mg/kg once daily efficacy [35]. Another, orally available PXS-4728A (also known as BI 1467335) dampened SSAO activity and blocked adhesion and tissue infiltration in patients with NASH (non-alcoholic steatohepatitis) (NCT03166735) [44], a condition that is also characterized by obesity [45]. Moreover, it was found to inhibit weight gain significantly in cholesterol-fed rabbits [26]. Wang et al. showed the efficacy of SSAO inhibition by novel hydrazine-containing small molecules [31]. Phenelzine, which is a derivative of hydrazine, is found to affect adiposity, glucose, insulin, and lipid homeostasis as well as markers of oxidative stress and low-grade inflammation in mice models. It represents a potent inhibitor of MAO and SSAO [40,46,47]. Holoamine 2-bromoethylamine (2-BEA) was also discovered to be a highly selective potential inhibitor of membrane-bound SSAO [30]. However, SSAO in fat is predominantly in soluble form. In addition, SSAO inhibition can have a positive influence not only on diabetes [48] but for some other pathologies, e.g., inflammation diseases [37]. Interestingly, Zinc-α2-glycoprotein (ZAG), a plasma protein with SSAO-inhibitive capacity [49], is found to reduce body weight drastically [50]. There are researches showing that SSAO substrate benzylamine ameliorates insulin secretion and glucose uptake in Goto-Kakizaki rats [24], dependent on the hydrogen-peroxide which is produced alongside SSAO activity. In this point of view, SSAO activity exerts anti-obesity and related alterations therapy value, e.g., diabetes. However, the mentioned process is accompanied by increasing oxidative stress and low-grade inflammation due to the elevation of cytotoxic compounds [51]. Thus, inhibiting SSAO reduces the risks of the complications caused by SSAO-mediated deamination products such as formaldehyde and hydrogen peroxide [52]. Remarkably, SSAO inhibition diminishes excessive fat deposition, thus, it ameliorates low-grade inflammation and reduces oxidative stress, which, on the other hand, prevents further complications [46,47].

## 4. Inhibition of SSAO by Caffeine

To sum up, there are studies which demonstrate the positive effects of SSAO-inhibitors on limiting excessive fat deposition, as well as ameliorating health condition in obese people via diminishing adipose accumulation [18]. Interestingly, some studies have revealed that imidazoline site ligands are the key compounds that can influence the inhibition of SSAO activity [53,54,55]. The present review proposes caffeine as an imidazole ring-containing substance which has the ability to bind SSAO inhibitory sites and inhibit the enzymatic activity [55]. Olivieri and Tipton have revealed the inhibitory concentration (IC) of caffeine intake—0.1–10 mM (IC_50_ = 0.8 ± 0.3 mM) [55], which roughly corresponds to 1–4 cups of regular coffee [56]. This amount is consistent with the recommended daily dose of caffeine (400 mg) for adults and is not associated with unfavorable effects on health [56]. However, it does not refer to children, pregnant women, or any vulnerable population. The illustration of an outcome of SSAO inhibition in obesity is given in Figure 1.

## 5. Dual Beneficial Role of Caffeine for Obese People

Caffeine is a heterocyclic organic compound, chemically known as 1,3,7-trimethylxanthine (C_8_H_10_N_4_O_2_) which represents xanthine consisting of a pyrimidine ring linked to an imidazole ring [57]. This neuro-stimulant commonly found in coffee beans, tea, cocoa, and chocolate is the most widely consumed natural alkaloid (Table 2). Caffeine enhances metabolism and is beneficial for digestion [58]. Consumption of coffee and tea where caffeine is abundantly present has shown weight loss and reduced risk of diabetes [59,60]. Westerterp-Plantenga et al. have studied caffeine influence on a randomized placebo-controlled double-blind parallel trial in 76 overweight and moderately obese subjects with habitual caffeine intake and green tea ingestion, and, consequently, the caffeine stimulated weight loss via thermogenesis and fat oxidation [10]. This result was corroborated by other studies [61,62]. Caffeine stimulates lipolysis via inhibiting the activity of phosphodiesterase (PDE) which degrades cyclic adenosine monophosphate (cAMP). Elevation of cAMP concentration activates the phosphorylation of hormone-sensitive lipase by protein kinase A. This process leads to lipolysis [63]. Moreover, Akiba et al. demonstrated that caffeine exerts inhibition of insulin-induced glucose uptake and also reduces GLUT4 translocation to the plasma membrane in the mouse pre-adipocyte cell line MC3T3-G2/PA6 [64]. Articles demonstrating caffeine-caused weight loss are presented in Table 3. An already well-known mechanism of caffeine-mediated weight loss is also associated with its capacity of adenosine receptors antagonism [65] Thus, it increases energy expenditure [66,67] and promotes alertness [68,69]. Since T2DM has increased over recent decades [70], caffeine, owing to the abovementioned abilities [71,72], might be a part of the treatment. Remarkably, NASH is firmly associated with obesity, as its main characteristic is liver fat accumulation in people who drink a little or no alcohol [45]. At the same time, SSAO levels are known to be elevated in the subjects with this medical condition [73,74] as well as overweight conditions [75]. Concomitantly, it is pivotal that coffee/caffeine consumption has been demonstrated to be able to reduce hepatic fibrosis in NASH patients significantly [76]. In different tissues, SSAO activity was diversely diminished after 10 days of caffeine administration, e.g., in the adipose tissue it was diminished by 41.4%, which was the highest amount compared to the rest of the samples (aorta, liver, kidney, serum) [13].

In a view of the fact that caffeine toxicity is low [83], it might be considered as a promising SSAO inhibitor which might aid the drug development in medicine against SSAO-mediated health disorders. Bakuradze et al. demonstrated that coffee as the major source of caffeine is favorable because of the beneficial properties against DNA oxidative damage with body fat reduction. Daily consumption of 3–4 cups of Arabica coffee is suggested for healthy subjects [80]. Neves et al. examined caffeine influence on mortality in women with diabetes and their findings showed the dose-dependent inverse impact on mortality [84].

To sum up, caffeine can be characterized as a valuable agent in two ways with the respect to weight loss: as an energy booster it enhances lipolysis and, via inhibiting SSAO, caffeine can be useful in the development of an anti-obesity therapy. Additionally, it can ameliorate the regulation of histamine levels while inhibiting SSAO which increases lipolysis in adipose tissue.

## 6. Does Caffeine Augment Histamine-Mediated Lipolysis in Adipose Tissue?

Histamine, a neurotransmitter which plays an important role in appetite regulation, seems to be an anorexigenic agent in terms of ameliorating leptin-resistance [85]. Histamine acts as a mediator for the inhibitory effect of leptin, a hunger-managing hormone which causes diminishing fat storage [86]. Histamine levels are downregulated in obese patients [85]. The enzyme that is responsible to catalyze histamine oxidation is diamine oxidase (DAO). Interestingly, in adipose tissue SSAO was found to oxidize this bioamine [87]. Histamine contains an imidazole ring which makes it more related to SSAO activity [88], especially in adipocytes where this enzyme is expressed abundantly and is highly active [87,89]. The first facts regarding histaminase inhibition by semicarbazide were published in 1942 [90]. This supports the idea that SSAO with increased activity is negatively related to histamine levels, meaning that histamine regulation might be a target of anti-obesity drug development. Evidently, SSAO activity is positively correlated to histamine oxidation [14]. As caffeine is able to inhibit SSAO as well as DAO [91], it may regulate levels of histamine which, eventually, stimulate fat lipolysis. John et al. studied caffeine influence on the activity of histamine in the posterior hypothalamus of rats and, remarkably, the release of histamine was found to be increased after caffeine administration [92]. Moreover, the existing data show the association of antihistamine drugs with gaining weight [82]. Yoshimoto et al. demonstrated that histamine receptor H3R agonist reduces adiposity in rodents [93]. Caffeine and histamine both contain imidazole rings which link them to SSAO activity. Histamine is presented in a number of plant foods such as eggplant, spinach, avocado, tomato [94], as well as in fermented dairy products: cheese, yogurt, whey, etc. [95]. Notably, dietary histidine, a precursor in histamine biosynthesis [96], is found to be an essential amino acid for maintaining histamine levels in the brain [97]. Histidine supplementation is also studied to improve insulin resistance in obese women with metabolic disorders [98]. Li et al. conducted a cross-sectional internet-based study on northern Chinese overweight adults and demonstrated that dietary histidine is inversely associated with obesity, insulin resistance, and inflammation [99]. Hence, histidine supplementation might influence SSAO levels and, therefore, limit body fat.

It is noteworthy that histamine and caffeine might cause certain health pathologies in high concentrations, e.g., human pregnancy complications [100], urticaria [101], and cardiovascular diseases [102] respectively. Thus, the right doses need to be applied.

## 7. Conclusions

Taken altogether, the reviewed literature suggests that the majority of the studies demonstrated the direct correlation between excessive fat accumulation and upregulated SSAO levels. Based on evidence, the agents able to inhibit SSAO seem promising for anti-obesity drug discovery, albeit some of them are acceptable while the others are characterized to have deleterious health effects. As the enzyme SSAO levels are increased in prediabetes, its inhibition might play an important role in diabetes prevention. In this review, we have stated the number of SSAO inhibitors, although all might not be applicable for humans due to their harmfulness. Nonetheless, caffeine as an agent capable to inhibit SSAO is assessed to be a promising option in terms of efficacy, low-toxicity, inexpensiveness, etc. The dual role of caffeine in anti-obesity includes the ability to reduce body fat through enhancing metabolism and to inhibit elevated SSAO activity. In addition, caffeine may indirectly augment histamine activity in adipose tissue and increase the speed of lipolysis. Histamine is hereby proposed to be a noteworthy substance to carry out the researches with respect to the enzyme SSAO, as it can be associated with obesity management. To conclude, habitual caffeine intake in safe doses may ameliorate SSAO levels and be efficient for the obese population.

## Figures and Tables

**Figure 1 nutrients-12-00184-f001:**
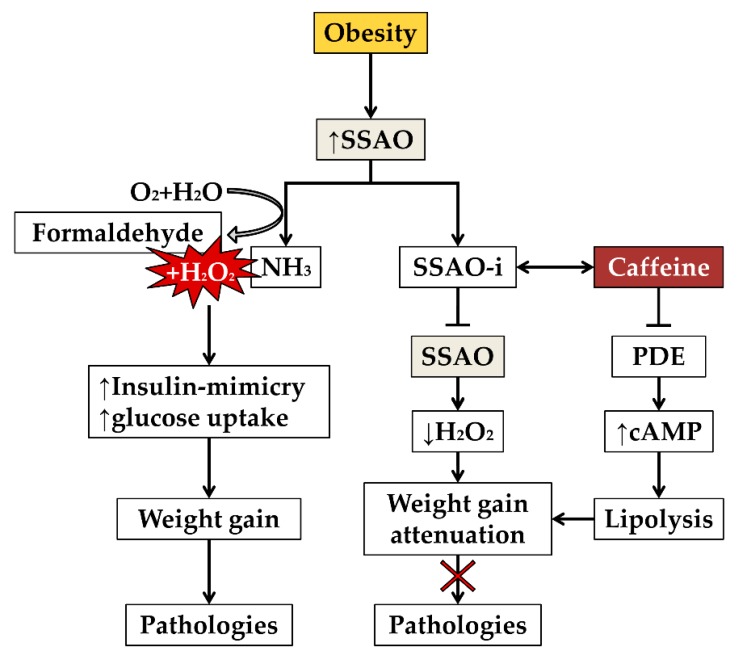
Illustration of SSAO involvement in weight gain and associated pathologies (on the left); beneficial health outcomes of SSAO inhibition and the dual mechanism of caffeine in diminishing weight gain (on the right). Notes: SSAO, semicarbazide-sensitive amine oxidase; SSAO-i, SSAO-inhibitor; PDE, phosphodiesterase; cAMP, cyclic adenosine monophosphate; ↑, upregulation; ↓, downregulation.

**Table 1 nutrients-12-00184-t001:** Description of semicarbazide-sensitive amine oxidase/vascular adhesion protein-1 (SSAO/VAP-1) inhibitors according to chemical and pharmacological properties.

Names of VAP-1/SSAO Inhibitors	Chemical Structure/Formula	Molecular Weight/Molar Weight	Solubility	Pharmacokinetic Profile	IC_50_	Efficacy/Anti-Obesity Property	Toxicity	Source
Oral Dose(Rat/Mouse)	i.v./i.p. Dose(Rat/Mouse)
PXS-4728A/BI1467335	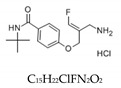	316.8014 kDa	DMSO and H_2_O >10 mg/mL	6 mg.kg^−1^10 mg.kg^−1^	3 mg.kg^−1^5 mg.kg^−1^	5 nM	Potent and orally available inhibitor of VAP-1, showing >500-fold selectivity for VAP-1/SSAO over all the related human amine oxidases. Diminishes lung inflammation. It is in clinical trials for the treatment of cardio-metabolic diseases. It shows significant reduction of body weight gain in rabbits.Axon Medchem, Groningen, Netherlands	No/Low	Wang et al. [26] Schilter et al. [28] Kim et al. [33]
PXS-4681A	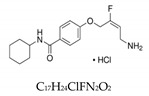	342.84 kDa	H_2_O 2 mg/mL	20 mg.kg^−1^2 mg.kg^−1^	10 mg.kg^−1^2 mg.kg^−1^	<10 nM	Potent and highly selective irreversible inhibitor of SSAO/VAP-1 that exhibits anti-inflammatory effects in vivo. It is a derivative of Mofegiline. PXS-4681A was used to inhibit LPS induced brain inflammation.Sigma-Aldrich, St. Louis, USA	No/Low	Becchi et al. [34] Foot et al. [35]
Semicarbazide	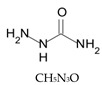	75.07 g/mol	N/A	N/A	N/A	N/A	An irreversible and probably suicide SSAO inhibitor. It limits weight gain and fat accumulation.Sigma-Aldrich, Saint Quentin Fallavier, France	Yes/High	Mercader et al. [22,36]
LJP-1586	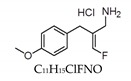	231.69 kDa	DMSO	10 mg/kg	N/A	4–43 nM	Potent, selective, and orally active inhibitor of SSAO activity, inhibiting vascular adhesion protein 1 (VAP-1) activity and decreasing the density of macrophages in inflamed atherosclerotic plaques in mice LJP.Glixx Laboratories Inc., Hopkinton, USA	Yes	O’Rourke et al. [37]
Caffeine	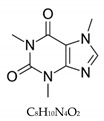	194.19 g/mol	H_2_O	N/A	N/A	0.8 ± 0.3 nM	Efficiency of caffeine on adipose and aorta is especially high. It can play an important role in treating diseases associated with SSAO activities. Independently from SSAO inhibition, it is found to be effective in losing weight.National Institute for Drug Control, Beijing, China	No/Low	Che et al. [13] Zheng et al. [38]
Simvastatin	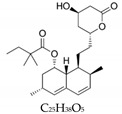	418.6 g/mol	DMSO and H_2_O	N/A	20 mg.kg^−1^	N/A	Simvastatin blocks SSAO/VAP-1 release, among other known actions, therefore preventing this cascade of events.Sigma-Aldrich, Madrid, Spain	Yes	Sun et al. [39]
Phenylhydrazine	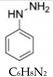	108.14 g/mol	H_2_O	N/A	N/A	30 nM	Irreversible SSAO inhibitor.Shows diminishing body weight gain.Sigma-Aldrich, Poole, UK	Yes/High	Carpene et al. [18] Lizcano et al. [29]
Phenelzine	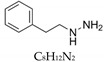	136.19 g/mol	H_2_O	30 mg.kg^−1^	88.9 µmol/kg	N/A	Potent inhibitor of SSAO.Shows attenuation of adiposity.Sigma-Aldrich, Saint Quentin Fallavier, France	Yes	Carpene et al. [40]

**Table 2 nutrients-12-00184-t002:** Caffeine content in popular food products.

Food and Beverages	Serving Size	Caffeine Content (mg)	Source
Coffee			Cappelletti et al. [77]
Instant	180 mL	90	
Instant decaffeinated		3	
Drip brewed		100	
Brewed decaffeinated		5	
Brewed	5 oz	135	Harland et al. [72]
Roasted and ground, drip		112	
Tea			Cappelletti et al. [77]
Green	180 mL	35	
Black		70	
Cocoa			
Cocoa beverages	180 mL	13	
Energy drinks			Kole et al. [78]
Red Bull	8.4 oz	80	
Sodas			Harland et al. [72]
Coca-Cola	12 oz	46	
Pepsi		38	
Chocolates			Nehlig [79]
Dark chocolate	50 g	35–200	
Milk chocolate		14	

**Table 3 nutrients-12-00184-t003:** Studies of caffeine-caused weight loss.

Title	Type of Study	Synergistic Effect	Impact of Weight Loss	Number of Subjects	Doses of Caffeine	Gender	Age	Summary	Source
Antioxidant-rich Coffee Reduces DNA Damage, Elevates Glutathione Status and Contributes to Weight Control: Results from an Intervention Study	Human	No	High	33	3–4 cups brewed	Male	20–44 y	3–4 cups of coffee daily reduces oxidative damage, body fat and ameliorates energy uptake	Bakuradze et al. [80]
Oral Intake of a Combination of Glucosyl Hesperidin and Caffeine Elicits an Antiobesity Effect in Healthy, Moderately Obese Subjects: a Randomized Double-blind Placebo-Controlled Trial	Human	Yes (Glucosyl-hesperidin)	High	75	50–75 mg	N/A	N/A	500 mg G-hesperidin and 75 mg caffeine together reduces body fat	Ohara et al. [81]
A Combination of Glucosyl Hesperidin and Caffeine Exhibits an Anti-obesity Effect by Inhibition of Hepatic Lipogenesis in Mice	Mice	Yes (Glucosyl-hesperidin)	High	N/A	N/A	Male	8 weeks	Caffeine + G-hesperidin effectively reduces body fat accumulation	Ohara et al. [82]
Caffeine Attenuated ER Stress-induced Leptin Resistance in Neurons	Cell culture	No	N/A	N/A	N/A	N/A	N/A	Caffeine may attenuate leptin resistance, thus, diminish obesity	Hosoi et al. [65]
Anti-obesity Effects of Three Major Components of Green Tea, Catechins, Caffeine and Theanine, in Mice	Mice	Yes (catechins)	High	100	N/A	Female	4 weeks	Caffeine + catechins suppress body weight and fat accumulation	Zheng et al. [38]
Anti-obesity Effect of a Novel Caffeine-Loaded Dissolving Microneedle Patch in High-fat Diet-induced Obese C57BL/6J Mice	Mice	N/A	High	N/A	N/A	Female	6 weeks	Novel caffeine loaded dissolving microneedle patch- CMP has therapeutic property in obesity	Dangol et al. [69]
Caffeine Inhibits Hypothalamic A1R to Excite Oxytocin Neuron and Ameliorate Dietary Obesity in Mice	Mice	No	High	N/A	60 mg.kg ^−1^	Male	6 weeks	Caffeine administration by central or peripheral route suppresses appetite, increases energy expenditure, and reduces the body weight	Wu et al. [66]
Effect of Chronic Coffee Consumption on Weight Gain and Glycaemia in a Mouse Model of Obesity and Type 2 Diabetes	Mice	No	High	N/A	N/A	Both	N/A	Regular coffee intake retards weight gain in high-fat diet mice and abolishes weight gain in normal diet mice	Rustenbeck et al. [60]
Caffeine Intake is Related to Successful Weight Loss Maintenance	Human	No	N/A	494/2129	1–7 cups of caffeinated beverages	Both	47.6/45.3 y	Consumption of caffeinated beverages might support weight loss maintenance	Icken et al. [11]
Body Weight Loss and Weight Maintenance in Relation to Habitual Caffeine Intake and Green Tea Supplementation	Human	Yes (epigallocatechin gallate)	High	76	150 mg/day	Both	18–68	Green tea-caffeine mixture intake ameliorates weight maintenance and weight loss	Westerterp-Plantenga et al. [10]

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
