# Peer review of "Attenuation of Weight Gain and Prevention of Associated Pathologies by Inhibiting SSAO"

_nutrients, 2020, doi:10.3390/nu12010184_

Round 1
Reviewer 1 Report
English language and style of this review article are required for minor spell check or editing. For example: "To replace “SSAO levels” with “SSAO activity” in line 20 To define VAP-1 and PrAO in the main text" .
In Figure1, section 4 and section 5 of the main text, the authors could include the major and detailed mechanisms of action of Caffeine for obese people.
SSAO activity in reducing obesity-related alterations should be highlighted in the main text.
How to explain the different results and impact on studies of caffeine-caused weight loss.
The author should explain the reason why SSAOi might not applicable for all obese humans.
Author Response
Dear Reviewer,
Thank you very much for reviewing and giving valuable comments and suggestions for our manuscript (Manuscript ID nutrients-679259) entitled “Attenuation of Weight Gain and Prevention of Associated Pathologies by Inhibiting SSAO”. Please find below a point-by-point reply to the reviewers’ concerns. We hope that you find our responses satisfactory. In the manuscript, we have highlighted all the added or changed text in red font color. Besides, in this cover letter, we have underlined the text copied from the manuscript.
First of all, as according to the journal “Nutrients” rules, co-first authors are not allowed, we removed the signs and the note of co-first authors on the first page of the manuscript. In addition, we have corrected some general misspelled mistakes. We also would like to inform you that the page numbers 17 have been changed into 18.
Comments to the Author:
Comment 1:
English language and style of this review article are required for minor spell check or editing. For example: "To replace “SSAO levels” with “SSAO activity” in line 20 To define VAP-1 and PrAO in the main text".
Answer: Thank you for your valuable comment.
We have changed “SSAO levels” to “SSAO activity” in line 18 (previously line 20) and defined VAP-1 (lines 31-32): It is noteworthy that an enzyme semicarbazide-sensitive amine oxidase (SSAO), also known as vascular adhesion protein-1 (VAP-1) which is responsible for deamination of the primary amines, such as methylamine, converts them into cytotoxic aldehydes, e.g. formaldehyde, ammonia, and hydrogen peroxide, is found to be associated with the obesity and related diseases; and PrAO (line 53) in the main text: SSAO is a copper-containing primary amine oxidase [15,16] (PrAO) and its activity is found to be increased in obesity.
Comment 2:
In Figure1, section 4 and section 5 of the main text, the authors could include the major and detailed mechanisms of action of Caffeine for obese people.
Answer: We are grateful for your helpful suggestions. We have taken your advice into consideration and added the major mechanism of action of caffeine in Figure 1 and its legend (lines 134-136; 137-138): Figure 1. Illustration of SSAO involvement in weight gain and associated pathologies (on the left); beneficial health outcomes of SSAO inhibition and dual mechanism of caffeine in diminishing weight gain (on the right).
Notes: SSAO, semicarbazide-sensitive amine oxidase; PDE, phosphodiesterase; cAMP, cyclic adenosine monophosphate; ↑, upregulation; ↓, downregulation; êž±, inhibition
Besides, we have added caffeine action mechanisms in section 5 (lines 149-152): Caffeine stimulates lipolysis via inhibiting the activity of phosphodiesterase (PDE) which degrades cyclic adenosine monophosphate (cAMP). Elevation of cAMP concentration activates the phosphorylation of hormone-sensitive lipase by protein kinase A. This process leads to lipolysis [64].
Comment 3:
SSAO activity in reducing obesity-related alterations should be highlighted in the main text.
Answer: Thank you very much for your comment. We have highlighted the SSAO inhibition effects on obesity-related alterations in section 3 (lines 116-118) Remarkably, SSAO inhibition diminishes excessive fat deposition, thus, ameliorates low-grade inflammation and reduces oxidative stress, which, on the other hand, prevents further complications [46,47].
Comment 4:
How to explain the different results and impact on studies of caffeine-caused weight loss.
Answer: Thank you for your valuable question. Although the results and impact of the articles demonstrating caffeine-caused weight loss are not exactly the same, all the mentioned studies in our manuscript are indicating positively in diminishing of weight gain. In addition, the types of studies are not the same for all the reviewed articles (meaning that some of them are humans, mice, or cell culture studies). Therefore, we presume that the main point through these articles is emphasized in our review.
Comment 5:
The author should explain the reason why SSAOi might not applicable for all obese humans.
Answer: Thank you for your suggestion. We have stated in section 7 that all the SSAOi might not be applicable for obese humans because of their toxicity (line 224-225): In this review, we have stated the number of SSAO inhibitors, although all might not be applicable for humans due to their harmfulness.

Reviewer 2 Report
Papukashvili and colleagues review the potential effect of SSAO inhibition on body weight management and propose caffeine as an effective, safe and reliable choice to inhibit SSAO activity. The review of this topic is rigorous and the author’s proposal is interesting since the toxicity of many SSAO inhibitors is high. In table 1, the authors show the chemical and pharmacological properties of several SSAO/VAP-1/PrAO inhibitors. The authors could consider to justify this selection. It would be logical to include in the table those SSAO inhibitors that have been particularly studied in the context of obesity, which is the topic of the present review. Among them, it is worth mentioning phenelzine because shows a stronger amine oxidase activity than other known SSAO inhibitors and, in addition, in normal-weight mice and diet-induced models of obesity phenelzine administration affects adiposity, glucose, insulin and lipid homeostasis, and markers of oxidative stress and low-grade inflammation (Carpéné C, Br J Pharmacol 2018; Carpéné C, Int J Mol Sci 2018; Mercader J, J Pharmacol Exp Ther 2019).
In lines 103-105 the authors mention that SSAO inhibition can have a positive influence on diabetes. As the sentence is written, it generates a conflict when considering SSAO substrates as an option to treat diabetes. It is convenient to disambiguate the SSAO activator and inhibitory effects related to hydrogen peroxide-dependent glucose uptake. Moreover, regarding positive influences of SSAO inhibition, its potential activity in reducing obesity-related alterations should be highlighted, particularly on oxidative stress and low-grade inflammation.
In table 3 or in the section 5 of the main text, when reviewing the potential caffeine mechanisms of action to reduce obesity, the authors could include the strong insulin-induced inhibition of glucose incorporation into lipids in isolated adipocytes.
The authors could consider the following changes:
To replace “SSAO levels” with “SSAO activity” in line 20 To define VAP-1 and PrAO in the main textAuthor Response
Dear Reviewer:
Thank you very much for reviewing and giving valuable comments and suggestions for our manuscript (Manuscript ID nutrients-679259) entitled “Attenuation of Weight Gain and Prevention of Associated Pathologies by Inhibiting SSAO”. Please find below a point-by-point reply to the reviewers’ concerns. We hope that you find our responses satisfactory. In the manuscript, we have highlighted all the added or changed text in red font color. Besides, in this cover letter, we have underlined the text copied from the manuscript.
First of all, as according to the journal “Nutrients” rules, co-first authors are not allowed, we removed the signs and the note of co-first authors on the first page of the manuscript. In addition, we have corrected some general misspelled mistakes. We also would like to inform you that the page numbers 17 have been changed into 18.
Comments to the Author:
Comment 1:
Papukashvili and colleagues review the potential effect of SSAO inhibition on body weight management and propose caffeine as an effective, safe and reliable choice to inhibit SSAO activity. The review of this topic is rigorous and the author’s proposal is interesting since the toxicity of many SSAO inhibitors is high.
In table 1, the authors show the chemical and pharmacological properties of several SSAO/VAP-1/PrAO inhibitors. The authors could consider to justify this selection. It would be logical to include in the table those SSAO inhibitors that have been particularly studied in the context of obesity, which is the topic of the present review. Among them, it is worth mentioning phenelzine because shows a stronger amine oxidase activity than other known SSAO inhibitors and, in addition, in normal-weight mice and diet-induced models of obesity phenelzine administration affects adiposity, glucose, insulin and lipid homeostasis, and markers of oxidative stress and low-grade inflammation (Carpéné C, Br J Pharmacol 2018; Carpéné C, Int J Mol Sci 2018; Mercader J, J Pharmacol Exp Ther 2019).
Answer: Thank you for your valuable comment. According to your suggestion, in table 1 and section 3 we have added some information of SSAO inhibitors regarding to obesity and included phenelzine in table 1 as well as in the text of section 3 (lines 102-105): Phenelzine, which is a derivative of hydrazine is found to affect adiposity, glucose, insulin, and lipid homeostasis as well as markers of oxidative stress and low-grade inflammation in mice models. It represents a potent inhibitor of MAO and SSAO [40,46,47].
Moreover, we have added information of another SSAO inhibitor Zinc-α2-glycoprotein (ZAG) (lines 108-109): Interestingly, Zinc-α2-glycoprotein (ZAG)- a plasma protein with SSAO inhibitive capacity [49] is found to reduce body weight drastically [50].
In addition, we have added weight-gain limiting property of PXS-4728A in the same section (lines 100-101) Moreover, it was studied to inhibit weight gain significantly in cholesterol-fed rabbits [26].
Comment 2:
In lines 103-105 the authors mention that SSAO inhibition can have a positive influence on diabetes. As the sentence is written, it generates a conflict when considering SSAO substrates as an option to treat diabetes. It is convenient to disambiguate the SSAO activator and inhibitory effects related to hydrogen peroxide-dependent glucose uptake.
Answer: Thank you very much for your important comment. As you suggested, we have justified the sentence (line 107-108, previously 103-105) by explaining that although some researches demonstrate the positive effects of SSAO substrates in anti-diabetic effects, other researches show the adverse effects of SSAO-mediated cytotoxic compounds. In the text of section 3 (lines 109-114): There are researches showing that SSAO substrate benzylamine ameliorates insulin secretion and glucose uptake in Goto-Kakizaki Rats [51] dependently on hydrogen-peroxide which is produced alongside with SSAO activity. In this point of view, SSAO activity exerts anti-obesity and related alterations therapy value, e.g. diabetes. However, the mentioned process is accompanied by increasing oxidative stress, low-grade inflammation due to the elevation of cytotoxic compounds [52].
Comment 3:
Moreover, regarding positive influences of SSAO inhibition, its potential activity in reducing obesity-related alterations should be highlighted, particularly on oxidative stress and low-grade inflammation.
Answer: Thank you for your valuable comment. According to your suggestion, we have added the SSAO inhibition effects on obesity-related alterations in section 3 (lines 116-118): Remarkably, SSAO inhibition diminishes excessive fat deposition, thus, ameliorates low-grade inflammation and reduces oxidative stress, which, on the other hand, prevents further complications [46,47].
Comment 4:
In table 3 or in the section 5 of the main text, when reviewing the potential caffeine mechanisms of action to reduce obesity, the authors could include the strong insulin-induced inhibition of glucose incorporation into lipids in isolated adipocytes.
Answer: We would like to thank you for the suggestion. We have included the mentioned information in section 5 (lines 151-153): Moreover, Akiba et al. demonstrated that caffeine exerts inhibition of insulin-induced glucose uptake and also reduces GLUT4 translocation to the plasma membrane in the mouse pre-adipocyte cell line- MC3T3-G2/PA6 [65].
Comment 5:
The authors could consider the following changes: To replace “SSAO levels” with “SSAO activity” in line 20 To define VAP-1 and PrAO in the main text
Answer: Thank you for your note. We have changed “SSAO levels” to “SSAO activity” in line 18 (previously line 20) and defined VAP-1 (lines 31-32): It is noteworthy that an enzyme semicarbazide-sensitive amine oxidase (SSAO), also known as vascular adhesion protein-1 (VAP-1) which is responsible for deamination of the primary amines, such as methylamine, converts them into cytotoxic aldehydes, e.g. formaldehyde, ammonia, and hydrogen peroxide, is found to be associated with the obesity and related diseases; and PrAO (line 53) in the main text: SSAO is a copper-containing primary amine oxidase [15,16] (PrAO) and its activity is found to be increased in obesity.
